# Interindividual Differences in In Vitro Human Intestinal Microbial Conversion of 3-Acetyl-DON and 15-Acetyl-DON

**DOI:** 10.3390/toxins14030199

**Published:** 2022-03-07

**Authors:** Fangfang Li, Jing Jin, Ivonne M. C. M. Rietjens, Fuguo Xing

**Affiliations:** 1Institute of Food Science and Technology, Chinese Academy of Agricultural Sciences/Key Laboratory of Agro-Products Quality and Safety Control in Storage and Transport Process, Ministry of Agriculture and Rural Affairs of P. R. China, 2 Yuanmingyuan West Road, Haidian District, Beijing 100193, China; l742240749@163.com (F.L.); jinjing@caas.cn (J.J.); 2Division of Toxicology, Wageningen University and Research, Stippeneng 4, 6708 WE Wageningen, The Netherlands; ivonne.rietjens@wur.nl

**Keywords:** acetyl-DONs, gut microbiota, deacetylation

## Abstract

In order to evaluate the potential differences between 3-Ac-DON and 15-Ac-DON in the human intestinal microbial metabolism, human fecal samples were anaerobically cultured in vitro. Quantitative fecal microbiota characteristics were obtained by 16S rRNA sequencing, and the data revealed several genera that may be relevant for the transformation of the acetylated DONs. Significant differences in the level of 3-Ac-DON and 15-Ac-DON conversion were observed among microbiota from different human individuals. 3-Ac-DON could be rapidly hydrolyzed; a ten-fold difference was observed between the highest and lowest in vitro conversion after 4 h. However, 15-Ac-DON was not fully transformed in the 4 h culture of all the individual samples. In all cases, the conversion rate of 3-Ac-DON was higher than that of 15-Ac-DON, and the conversion rate of 3-Ac-DON into DON varied from 1.3- to 8.4-fold that of 15-Ac-DON. Based on in vitro conversion rates, it was estimated that 45–452 min is required to convert all 3-Ac-DON to DON, implying that deacetylation of 3-Ac-DON is likely to occur completely in all human individuals during intestinal transit. However, for conversion of 15-Ac-DON, DON formation was undetectable at 4 h incubation in 8 out of the 25 human samples, while for 7 of these 8 samples conversion to DON was detected at 24 h incubation. The conversion rates obtained for these seven samples indicated that it would take 1925–4805 min to convert all 15-Ac-DON to DON, while the other 17 samples required 173–734 min. From these results it followed that for eight of the 25 individuals, conversion of 15-Ac-DON to DON was estimated to be incomplete during the 1848 min intestinal transit time. The results thus indicate substantial interindividual as well as compound specific differences in the deconjugation of acetylated DONs. A spearman correlation analysis showed a statistically significant relationship between deconjugation of both acetyl-DONs at 4 h and 24 h incubation. Based on the in vitro kinetic parameters and their scaling to the in vivo situation, it was concluded that for a substantial number of human individuals the deconjugation of 15-Ac-DON may not be complete upon intestinal transit.

## 1. Introduction

*Fusarium* fungi is a worldwide producer of a range of mycotoxins. Deoxynivalenol (DON), which belongs to group B trichothecenes, is one of the most prevalent food-associated mycotoxins mainly produced by *Fusarium graminearum* and *Fusarium culmorum*, frequently contaminates cereals and cereal products [1]. In addition to DON, cereals and feed are often co-contaminated with their acetylated derivatives 3-acetyl-DON (3-Ac-DON), 15-acetyl-DON (15-Ac-DON) and its glycoside (DON-3-Glc) [2]. A survey of 13,629 samples from 77 countries around the world showed that the pollution rates for each continent were as follows: North America, 67%; South and Central America, 67%; Europe, 63%; Asia, 80%; the Middle East, 65%, and Africa, 75%, suggesting regular human exposure [3]. Global scientists pay attention to DON and its derivatives because of their high contaminated rates and significant deleterious effects on humans and animals [2,4], and the fact that risk assessments by food safety authorities reported exposure to potentially exceed established health-based guidance values for parts of the population [5,6]. The occurrence of DON was observed in 450 wheat samples collected during 2013–2015 in Jiangsu province, China. 148 out of 450 samples (98.7%) were contaminated with DON levels ranging up to 18,709.4 μg/kg (mean 1628.6 μg/kg) in 2015. The average concentrations of DON were 879.3 ± 1127.8, 627.8 ± 640.5, and 1628.6 ± 2168.0 μg/kg in 2013–2015, respectively [7].

In addition to its high prevalence, DON is one of the most hazardous food or feed associated mycotoxins [8]. DON is known to cause a variety adverse effects upon acute exposure, of which much is related to the gastrointestinal tract. The acute effects include, among others, nausea, vomiting, gastrointestinal upset, dizziness, diarrhea and headache [1]. LD50 for mice ranges from 49 to 70 mg/kg (intraperitoneal DON injection), and 46 to 78 mg/kg (oral DON administration). LD50 for a 10-day old duckling is 27 mg/kg when the toxin is administered subcutaneously and 140 mg/kg for one-day-old broiler chicks with DON oral administration [9]. DON also causes human and animal emesis, anorexia and growth suppression, affects intestinal absorption of nutrients, increases susceptibility to infections, and causes chronic diseases [2]. Similar to their parent compound DON, it was reported that the two derivatives 3-Ac-DON and 15-Ac-DON cause intestinal function toxicity and immunotoxicity [2]. Several cytotoxic assays reported that 3-Ac-DON was more toxic than 15-Ac-DON [1], whereas other studies presented opposite views, with 15-Ac-DON found to be higher toxic than 3-Ac-DON and DON [10], so that the relative toxicity of acetylated DONs appears to depend on the type of determination and the endpoint of the toxicity [2].

Previous studies have shown that the gut microbiota of different host species, such as cattle, pig, chicken, and humans, can uncouple modified DON from DON, resulting in increased exposure of DON [2,11]. Young et al. [12] proved that microbial bacteria and pure cultures from chicken bowel can degrade two acetylated DONs and DON by the way of deacetylation and/or oxidation. Gratz et al. [13] confirmed that in in vitro experiments, human fecal microbial bacteria could transform glycosylated form D3G to parent form DON, and further DON was detoxicated to DOM-1. Eriksen and Pettersson [14] incubated 3-Ac-DON with human feces under anaerobic conditions. The results obtained showed that only DON was detected, while under the experimental conditions applied formation of DOM-1 was not detected. The modified DON can be unbound after ingestion, and this transformation of human and animal intestinal microbial bacteria increased the exposure of DON and the risk caused by this mycotoxin.

This study aims to gain insight into potential interindividual differences in gut microbial bacteria transformation of acetylated DONs and to evaluate the results within the framework of acetylated DONs risk assessment. In addition, fecal microbial bacteria taxonomic profiles were quantitatively characterized by 16S rRNA sequencing analysis to determine correlations between kinetic parameters and taxon abundances in different human fecal samples.

## 2. Results

### 2.1. Quantitative Bacterial Taxonomy Profiling Analysis

Metagenomics analysis was carried out on 25 individual human fecal samples. Compared with the SILVA 138 (http://www.arb-silva.de/, access date: 3 May 2020) SSUrRNA database, 874 Operational Taxonomics Units (OTUs) were annotated, and 63 core OTUs shared by all the samples were identified (Figure 1). The relative microbial profile at the phylum level of the 25 individual fecal samples is reported in Appendix A. The bacteria belong to ten different phyla, including *Firmicutes*, *Proteobacteria*, *Actinobacteriota*, *Bacteroidota*, *unidentified_Bacteria*, *Fusobacteriota*, *Euryarchaeota*, *Desulfobacterota*, *Verrucomicrobiota*, and *Cyanobacteria*. Interindividual differences among human samples in the microbial relative abundance were observed. Microbial bacteria belonging to *Firmicutes*, *Proteobacteria* and *Actinobacteriota* were found to be the top three dominant phyla in all samples. 

At the genus level, the relative bacterial taxonomy profile of the 25 samples was reported (Figure 2). Substantial differences were observed when comparing the most abundant genera of different humans. The top ten dominant genera were *Bacteroides*, *Bifidobacterium*, *Vulcaniibacterium*, *Prevotella*, *Faecalibacterium*, *Megamonas*, *Schlegelella*, *Blautia*, *Streptococcus*, and *Collinsella*. The top 35 genera with the highest abundance were selected according to the annotation and abundance information of all samples. A heat map was drawn to visualize the hierarchic clustering of different individuals based on the genus level (Figure 3). At the species level, the top ten were *Alcanivorax_venustensis*, *Romboutsia_ilealis*, *Bacteroides_uniformis*, *Ruminococcus_sp_N15.MGS-57*, *Fusobacterium-mortiferum*, *Anoxybacillus_flavithermus*, *Bifidobactrium_adolescentis*, *Collinsella_aerofaciens*, *Bacteroides_vulgatus*, and *Faecalibacterium_prausnitzii* (Figure 4).

### 2.2. Interindividual Differences of Human Gut Microbial Conversion of 3-Ac-DON and 15-Ac-DON

Incubation periods of 4 h and 24 h were chosen to quantify potential human interindividual differences in the intestinal microbial metabolism of the acetyl-DONs. Both tested mycotoxins and also DON itself were found to be stable during control incubations without feces (data not shown). At 200 μM substrate concentration, after 4 h incubation in the presence of 33.3 mg/mL feces, conversion of 3-Ac-DON to DON was observed in anaerobic incubations with all 25 samples, albeit to a different extent varying between 1.8 to 17.8% of the original parent compound of 3-Ac-DON added (Figure 5A). Thus, upon 4 h incubation a 10-fold difference in the deacetylation rate of 3-Ac-DON was observed between the highest and lowest in vitro conversion for the 25 individual fecal samples. The microbial deacetylation of 3-Ac-DON continued in a time dependent manner so that at 24 h, incubation conversion of 3-Ac-DON to DON varied between 4.5 and 54.0% of the starting amount of 3-Ac-DON added (Figure 6A). Under the conditions applied in the incubation experiments, no other metabolites were detected. After 24 h incubation, a 11.9-fold difference was observed between the highest and lowest conversion. Fecal microbial conversion of 15-Ac-DON to DON by the 25 individuals after 4 h incubation is shown in Figure 5B. The fecal sample from individual H9 showed the highest conversion rate, with up to 9.26 ± 1.21 µM DON formed (4.6% conversion) upon 4 h incubation. The results in Figure 5B also reveal that at 4 h anaerobic incubation with fecal samples of about one third (eight individual samples) of the 25 individual samples, no formation of DON was detected. After 24 h, the highest level of DON formation was 18.06 ± 0.38 µM (9.0% conversion) observed for the fecal sample from individual H18. At 24 h of incubation there was still one individual sample (H1) that showed no detectable DON formation (Figure 6B). Under the conditions applied in the incubation experiments, no other metabolites were detected.

### 2.3. Estimation of Microbial Transformation of 3-Ac-DON and 15-Ac-DON by Human Individuals In Vivo

Under the conditions applied, the conversion of 13-Ac-DON and 15-Ac-DON by human fecal samples had already been shown previously to be linear in time and with the amount of fecal sample [11], enabling in vivo estimation of the corresponding conversion rates. A time period of 4 h was chosen to quantify potential inter-human differences in gut microbial metabolism of 3-Ac-DON and 15-Ac-DON. From the data obtained at 4 h incubation, rates for the conversion of two acetylated DONs into DON in incubations with individual human feces were determined, except for the 8 individuals for which conversion of 15-Ac-DON to DON was not detectable at 4 h for which the data at 24 h were used. The values thus obtained are presented in Table 1. The in vitro transformation rates for 3-Ac-DON amounted to 0.44 to 4.42 nmol/min/g of feces at 200 μM acetylated substrate. In the following step, the rates were used to estimate the time required to convert 100% of the 200 μM acetylated DONs. Assuming one ml equals one gram of feces, it would take 45–452 min to convert all 3-Ac-DON to DON for the 25 individual human samples. Given that the reported intestinal transit time in humans is 1848 min for Asian people [15], it can be concluded that the deacetylation of 3-Ac-DON is likely to be complete in all the human individuals. For conversion of 15-Ac-DON, eight out of the 25 human samples appeared not to generate detectable levels of DON formation at 4 h incubation. For the remaining 17 samples, it would take 173–734 min to convert all 15-Ac-DON to DON. According to the data obtained at 24 h incubations, for seven of the eight individuals for which DON formation was not detected at 4 h incubation, DON formation at 24 h indicated 0.04 to 0.10 nmol/min/g of feces transformation rates. The fecal sample of one individual appeared unable to deconjugate 15-Ac-DON even upon 24 h incubation time. For the other seven it was calculated that it would take about 2000–5000 min to convert all 15-Ac-DON to DON (Table 1). Considering the reported human transit time (1848 min), these eight human individual samples may not be able to transform all the 15-Ac-DON during the available transit time. Based on the results obtained it thus appears that the level of 15-Ac-DON deconjugation to DON would vary from 0 to 100%, pointing at substantial interindividual differences (Table 1). The results also point at an important difference in the deconjugation efficiency for 3-Ac-DON and 15-Ac-DON. The hydrolysis of 15-Ac-DON by human fecal samples was obviously slower than that of 3-Ac-DON, resulting in incomplete hydrolysis of 15-Ac-DON to DON upon intestinal transit for some individuals. A spearman correlation analysis showed a statistically significant relationship between deconjugation of both acetyl-DONs by the 25 individual samples both at 4 h (r^2^ = 0.657, *p* < 0.01) and at 24 h incubation (r^2^ = 0.730, *p* < 0.01) (Figure 7A,B). The slope of the correlation amounted to 0.21 and 0.17 indicating the deconjugation of 15-Ac-DON to be 4.8 to 5.9 fold less efficient than that of 3-Ac-DON.

### 2.4. Correlation Analysis between Intestinal Microbial Profile and Metabolite Formation

In order to elucidate potential correlations between the microbial taxonomic characteristics of the 25 human fecal samples and their activities for 3-Ac-DON or 15-Ac-DON deacetylation, a Spearman correlation study was conducted between the relative abundance of the bacteria at species level and the concentrations of DON formed after 4 h and 24 h cultivation in the fecal slurry incubations. A heatmap shows the Spearman correlation coefficients (Figure 8). The results obtained revealed several significant correlations (*p* < 0.05 or *p* < 0.01) for DON formation from 3-Ac-DON and 15-Ac-DON and the differential abundance of microbial genera. The rate of DON formation from 3-Ac-DON, for example, showed positive correlation with the amounts of species *Ruminococcus_sp_N15.MGS.57* and *Bacteroides_uniformis*. The DON formation form 15-Ac-DON showed positive correlation with the amount of species *Fusobacterium_mortiferum* and *Anoxybacillus_Flavithermus*.

## 3. Discussion

The aim of this study was to assess the interindividual differences in the intestinal microbial transformation of 3-Ac-DON and 15-Ac-DON to DON using in vitro anaerobic fecal incubations. Acetylated DONs are major modified forms of DON present in food and feed, and due to their deacetylation by human intestinal microbiota [11], the exposure of acetylated DONs adds to the overall DON exposure. Assuming even distribution of a dose level over the large intestinal content of 2.06 L [16], an overall concentration of 200 µM would result from a dose level of 1991 µg kg^−1^ b.w. in humans. The dose level we used in the incubation experiments is close to the level of DON detected in food. The results obtained in the present study confirm that the human intestinal microbiome can transform acetylated DONs to DON, while under the current experimental conditions, further conversion of DON to DOM-1 was not observed, the latter in contrast to what was reported in comparable studies for other species [12,17]. In the present study, substantial human interindividual differences in transformation of the two acetylated DONs were observed.

In the present study, an in vitro anaerobic incubation method was applied to investigate interindividual differences in human intestinal microbial conversion of 3-Ac-DON and 15-Ac-DON. Differences exist in bacterial numbers and compositions along the intestinal tract which might lead to regiospecific differences. Colon is the main site for bacterial fermentation, as it harbors 70% of the total bacteria present in the gut [18]. The bacterial communities in colon and feces have been reported to be highly comparable, thus the use of anaerobic fecal samples as a representative population of intestinal microbes [19,20]. The use of fecal samples offers a number of advantages for the study of intestinal microbial metabolism of acetylated DONs and other xenobiotics. The use of fecal samples provides an easy way to study interindividual differences in gut microbiota mediated metabolism. The samples are obtained non-invasively at sufficiently high amounts, thereby providing the possibility for non-invasive studies on human volunteers. These in vitro incubation studies enabled the quantification of metabolic rates for the deacetylation of the acetylated DONs. Previous studies already provided proofs of principle for the adequacy of using anaerobic in vitro incubations with fecal samples for quantification of kinetics for intestinal microbial metabolism [16,21]. 

The observation of conversion of acetylated DONs to DON by intestinal microbiota is in accordance with previous reports. In our previous study [11], deacetylation of 3-Ac-DON and 15-Ac-DON by a pooled human fecal sample from 10 healthy volunteers was reported. Based on deconjugation rates obtained with this pooled human fecal sample it was concluded that 3-Ac-DON and 15-Ac-DON can be fully deconjugated upon intestinal transit. In the present study, it was shown that, in spite of a 10-fold interindividual difference in the deacetylation rate for 3-Ac-DON, 3-Ac-DON is also likely to be fully deconjugated within the reported intestinal transit time at the individual level. However, interindividual differences in 15-Ac-DON deconjugation were also substantial, and even such that it appeared that 15-Ac-DON may not be fully transformed upon intestinal transit by all individuals. For eight out of the 25 individual samples, deconjugation of 15-Ac-DON upon intestinal transit was estimated to be incomplete. This discrepancy may be due to the fact that the volunteer’s age, gender, race, and dietary habits may influence the human microbiome, potentially resulting in variable rates for deconjugation of acetylated DONs.

Eriksen et al. [14] reported that 48 h incubation of 3-Ac-DON with human fecal samples (1 mL feces suspended in 9 mL degassed McDougal buffer solution) resulted in 78% ± 30% deacetylation of 3-Ac-DON. Ajandouz et al. [22] reported that fecal incubations with human samples (obtained from three male volunteers, 45–50 years old, 1 mL feces suspended in 9 mL degassed PBS buffer solution) showed 3-Ac-DON deacetylation activity (22.6% ± 15.9%) and no activity on 15-Ac-DON after 6 h incubation. When incubation time was up to 48 h, 82.1% ± 21.9% and 14.9% ± 3.2% deacetylation for 3-Ac-DON and 15-Ac-DON appeared, respectively. The calculated 3-Ac-DON deacetylation rate was reported to amount to 37 ± 1.6 pmol/min/mg of feces based on DON formation, and for 15-Ac-DON it was 14 ± 0.6 pmol/min/mg feces. Based on these deacetylation rates, this study reported deacetylation of 3-Ac-DON to be 2.64 fold more efficient than that of 15-Ac-DON, a difference in line with our result. In the present study, we additionally revealed substantial interindividual and compound specific differences in the ability to deconjugate the two acetylated DONs among the 25 volunteers. 

The human gut microbiota plays a key role in the metabolism of xenobiotics, transforming hundreds of dietary components into metabolites that may display different activities, toxicities and kinetics in the human body. Molecular level understanding of gut microbial xenobiotic metabolism will guide personalized toxicology risk assessment [23]. As far as we know, the gut microbes involved in mycotoxin metabolism are poorly understood. In this study, we revealed that *Ruminococcus_sp_N15.MGS.57* and *Bacteroides_uniformis* showed a positive correlation with the rate of 3-Ac-DON deacetylation, and *Fusobacterium_mortiferum* and *Anoxybacillus_Flavithermus* were positively correlated with deacetylation of 15-Ac-DON. In a previous study, we suggested the *Lachnospiraceae* as a key contributor to the de-acetylation of acetyl-DONs of different individuals and species (mice, rat, pig and human) [11]. Daud et al. [24] proved that *Bacteroidaceae* and *Lachnospiraceae* can effectively de-acetylate acetylated forms of T-2 toxin and 4,15-diacetoxyscirpenol. The aforementioned studies indicate that several different gut microbiome genera may be involved in the deacetylation of acetyl-DONs. The variation of intestinal microbial communities among individuals has an effect on mycotoxin metabolism. The spearman correlation analysis showed a statistically significant relationship between acetyl-DONs (r^2^ = 0.657 (4 h), 0.730 (24 h), *p* < 0.01) transformed by the 25 individual samples, suggesting that in part the same gut bacteria may be involved in the deacetylation process of both acetyl-DONs.

## 4. Conclusions

In conclusion, we characterized substantial interindividual and also compound specific differences in the transformation of acetylated DONs by the human intestinal microbiome present in fecal samples. By quantifying the kinetics and comparison of the conversion rates with the reported asian people’s intestinal transit time, the in vitro data can be used to evaluate the in vivo situation in humans. The deacetylation of 3-Ac-DON by human fecal flora was significantly (max 8.4-fold, on average 4.8 to 5.9 fold) faster than that of 15-Ac-DON, and in contrast to deacetylation of 3-Ac-DON the deacetylation of 15-Ac-DON to DON upon intestinal transit was shown to not likely be complete for a substantial number of human individuals. Correlation analysis indicated that some gut microbial bacteria genera may be involved in the deacetylation of acetyl-DONs, while there may also be overlap in the gut bacteria involved in the 3-Ac-DON and 15-Ac-DON deconjugation process. It is concluded that interindividual differences in deacetylation of acetylated DONs exist, but that in risk assessment assumption of complete intestinal deconjugation of 3-Ac-DON provides an adequate approach.

## 5. Materials and Methods

### 5.1. Chemicals and Solutions

The mycotoxins DON, 3-Ac-DON and 15-Ac-DON were bought from FERMENTEK ltd. (Jerusalem, Israel). DOM-1 in acetonitrile was purchased from Sigma-Aldrich (Burlingtong, MA, USA). DMSO was purchased from Sigma-Aldrich (Burlingtong, MA, USA). Phosphate buffered saline (PBS) was bought from Thermo Fisher (Waltham, MA, USA). Chromatography grade acetonitrile was bought from Romer Labs (Tullin, Austria).

DON, 3-Ac-DON and 15-Ac-DON were dissolved in DMSO, final concentrations of the stock solutions were 100 mM, all standard solutions were stored at −20 °C. 

### 5.2. Human Fecal Slurry Preparation

Human fecal samples in this study were obtained from 25 healthy volunteers (19 women and six men), and their ages were between 24–30 years old. Individual information is provided in the Appendix A. All volunteer donors submitted a standard questionnaire to collect information related to their health status, lifestyle and drug use. The selected donors were healthy and without any gastrointestinal diseases. They ate a regular diet and did not receive any antibiotic treatment or probiotic preparations in the previous six months. The study was exempted and approved by the Medical Ethical Review Board of the Institute of Food Science and Technology, Chinese Academy of Agricultural Sciences. All donors knew the purpose and protocol of this study and submitted their written consent. The fecal samples were usually collected at 8:00 a.m. on the collection days, and were not polluted by urine. 3–5 g of fecal samples were collected in specimen tubes and subsequently transferred into an anaerobic environment within 5 min after donation by the participants.

Fecal samples were collected immediately after donation, weighted, and then transported within 30 min to the anaerobic condition by using a BACTRON 300 operating chamber (Cornelius, WA, USA). The condition of the atmosphere in the chamber was 85% N_2_, 10% CO_2_, and 5% H_2_, at 37 °C. Fecal samples were diluted fivefold with a sterile PBS buffer (containing 10% glycerol). After mixing, fecal slurries were manually homogenized using a vortex. After that, the slurries were filtered by using an eight layer sterile medical gauze swab under anaerobic conditions. Aliquots of the slurries were frozen and stored at −80 °C until further use.

### 5.3. Fecal Batch Culture Incubations

To characterize the microbial transformation of acetyl-DONs by intestinal microbiota, incubations with individual fecal samples were performed. The incubation mixtures were 1000 μL, prepared in de-oxygenated PBS. The incubation system contained either 10 μL 20 mM DON, or acetylated DONs (final concentration 200 μM) and 200 μL five times sterile PBS diluted feces (final fecal concentration 3.3%, i.e., 33.3 mg faces/mL). Aliquots of incubated fecal samples (200 µL) were collected after 0, 4 and 24 h incubation. The incubation procedure was terminated by adding the same volume of ice-cold methanol to 200 µL incubation samples and keeping the samples on ice for 30 min. After the reaction, the samples were centrifuged at 4 °C, 21,500× *g* for 20 min to precipitate particles and proteins, etc. The supernatant of each sample was transferred to a clean eppendorf tube and kept at −80 °C for further analysis by HPLC. Control groups were blank incubations without feces. All of the incubations were performed three times.

### 5.4. HPLC Analysis of Fecal Incubation Samples

The concentration of mycotoxins (DON and two acetylated DONs) and their metabolites in the supernatant from incubation samples was analyzed and quantified. The mycotoxin stock solutions were diluted in acetonitrile and freshly prepared. Calibration curves were built from stock solutions by creating nine standards as follows: 1, 2.5, 5, 10, 25, 50, 100, 250, and 500 µM. The analysis was undertaken using an Agilent 1220 Infinity II system connected to a photodiode array detector (Agilent Technologies, Palo Alto, California, CA, USA). The column was an Agilent TC-C18 (250 mm × 4.6 mm). The flow rate was 1.0 mL/min. 20 μL of each sample was injected with a mobile phase composed of nanopure water (A) and acetonitrile (B). The gradient elution program was as follows: starting condition 100% A, 1.0 min 20% A, 20.0 min 20% A, 25.0 min 100% A, 30.0 min 100% A. The concentration of mycotoxins and their metabolites was obtained by comparing the peak areas to the corresponding built calibration curves at a wavelength of 237 nm. 

### 5.5. Microbial Taxonomic Profiling

Frozen aliquots of the fecal samples of the 25 donors were analyzed by Novogene (Beijing, China). DNA were extracted with the QIAamp DNA stool mini kit (Qiagen, Valencia, CA, USA) following the protocol provided by the supplier. Extracted genomic DNA (2 ng/μL) was used for library preparation. The purity and integrity of the DNA was determined with a nanodrop (ND-1000) spectrophotometer (Nanodrop Technologies, Wilmington, DE, USA) through 1% agarose gel electrophoresis (AGE). DNA concentration was measured using Qubit^®^ dsDNA Assay Kit in Qubit^®^ 2.0 Fluorometer (Carlsbad, CA, USA). The OD value was between 1.8–2.0, and DNA contents above 1μg were used to construct the library. DNA samples were amplified using 16S V3−V4 primers. Specific forward and reverse primers (F-NXT-Bakt-341F: 5′-CCTACGGGNGGCWGCAG-3′ andR-NXT-Bakt-805R: 5′-GACTACHVGGGTATCTAATCC-3′) were used in the PCR reactions procedure. Amplifications were performed in 25 µL reactions with Qiagen HotStar *Taq* master mix (Qiagen Inc, Valencia, CA, USA), 2 µL of primers, and 1 µL of template. Raw sequence data from the Illumina NovaSeq 6000 (Illumina^®^ Inc. Columbia Circle, Albany, NY, USA) platform was processed, and 250 bp paired-end reads were generated. Microbiota identification was carried out by clustering the operational taxonomic units (OTU) from phylum to genus level using MUSCLE (Version 3.8.31) software.

### 5.6. Data Analysis

The results are shown as mean ± standard deviation (SD). Statistical significance was tested with a Student’s *t*-test (*p* < 0.05). The degradation percentage of the mycotoxins was calculated as follows:D_r_ = (1 – C_t_/C_ck_) × 100%(1)
where D_r_ represents the degradation percentage at a certain incubation time, C_t_ is the concentration of the metabolite of interest (DON), and C_ck_ is the concentration of parent compounds (3-Ac-DON, 15-Ac-DON) in Time 0 [23]. The origin version 2019b (Northampton, MA, USA) was used to process and statistically analyze the data obtained by HPLC. SPSS Statistics 26.0 (IBM, Chicago, IL, USA) was used to calculate the Spearman’s rank correlation coefficients *p* (−1 ≤ *p* ≤ 1) of the rate of deconjugation of two acetylated DONs after 4 h or 24 h incubation.

## Figures and Tables

**Figure 1 toxins-14-00199-f001:**
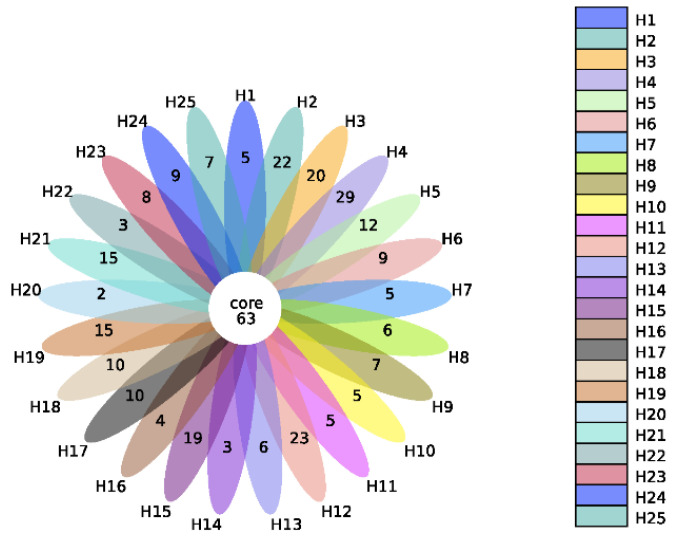
Flower figure of the number of OTUs and the overlap of OTUs (Alpha Diversity) for the different individual human fecal samples (each color represents one human sample).

**Figure 2 toxins-14-00199-f002:**
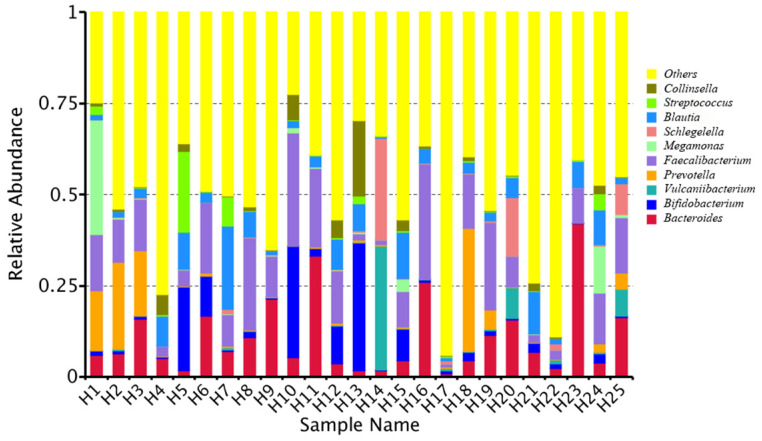
Analysis of the microbiome relative abundance at genus level in individual human samples.

**Figure 3 toxins-14-00199-f003:**
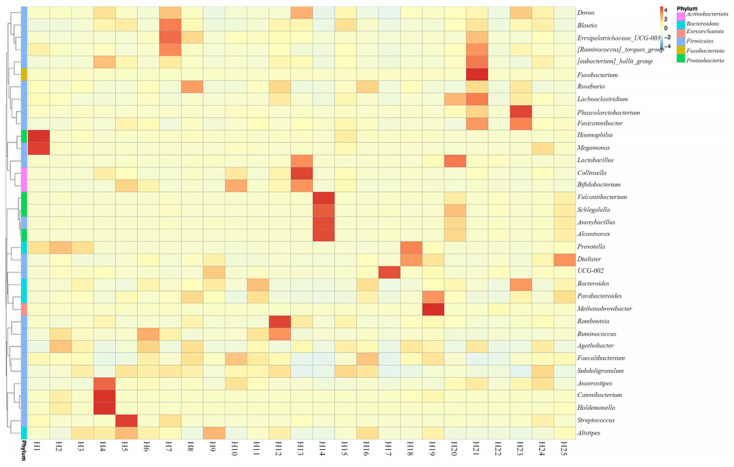
Heatmap depicting the differential abundance of microbial genera between individual human samples (*p*  <  0.01).

**Figure 4 toxins-14-00199-f004:**
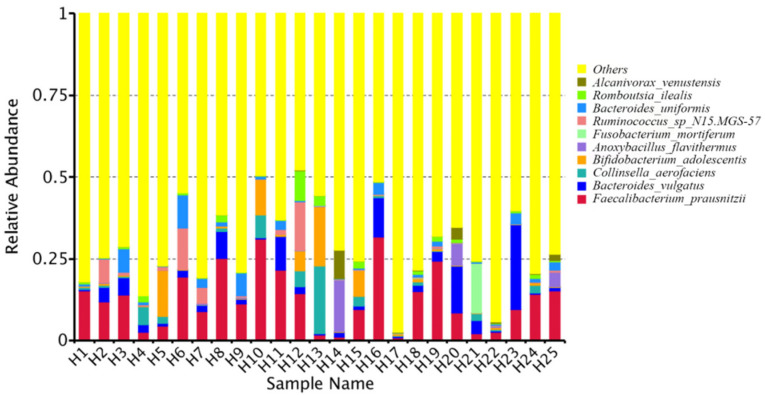
Analysis of the microbiome relative abundance at species level in individual human samples.

**Figure 5 toxins-14-00199-f005:**
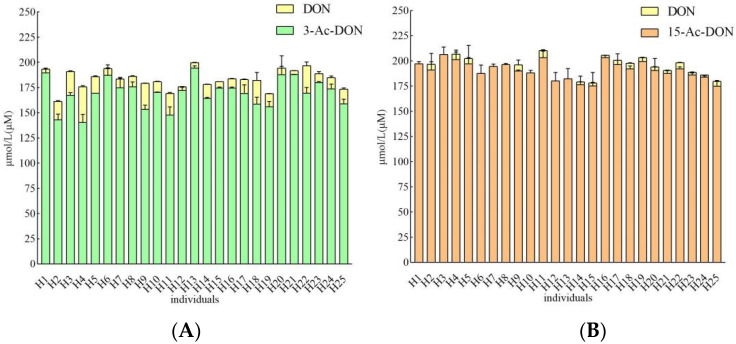
(**A**). Metabolism of 200 µM 3-Ac-DON by individual human fecal microbiota at 4 h. Results are presented as average ± SD of three independent experiments. (**B**). Metabolism of 200 µM 15-Ac-DON by individual human fecal microbiota at 4 h. Results are presented as average ± SD of three independent experiments.

**Figure 6 toxins-14-00199-f006:**
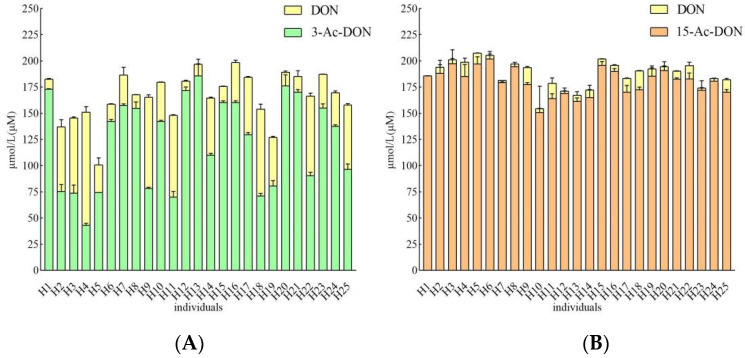
(**A**). Metabolism of 200 µM 3-Ac-DON by individual human fecal microbiota at 24 h. Results are presented as average ± SD of three independent experiments. (**B**). Metabolism of 200 µM 15-Ac-DON by individual human fecal microbiota at 24 h. Results are presented as average ± SD of three independent experiments.

**Figure 7 toxins-14-00199-f007:**
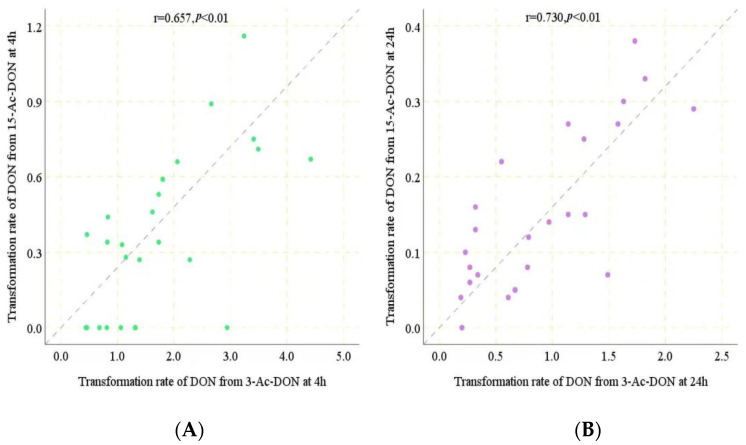
(**A**). Spearman correlation analysis between transformation rate of DON from 3-Ac-DON and 15-Ac-DON at 4 h incubation. (**B**). Spearman correlation analysis between transformation rate of DON from 3-Ac-DON and 15-Ac-DON at 24 h incubation.

**Figure 8 toxins-14-00199-f008:**
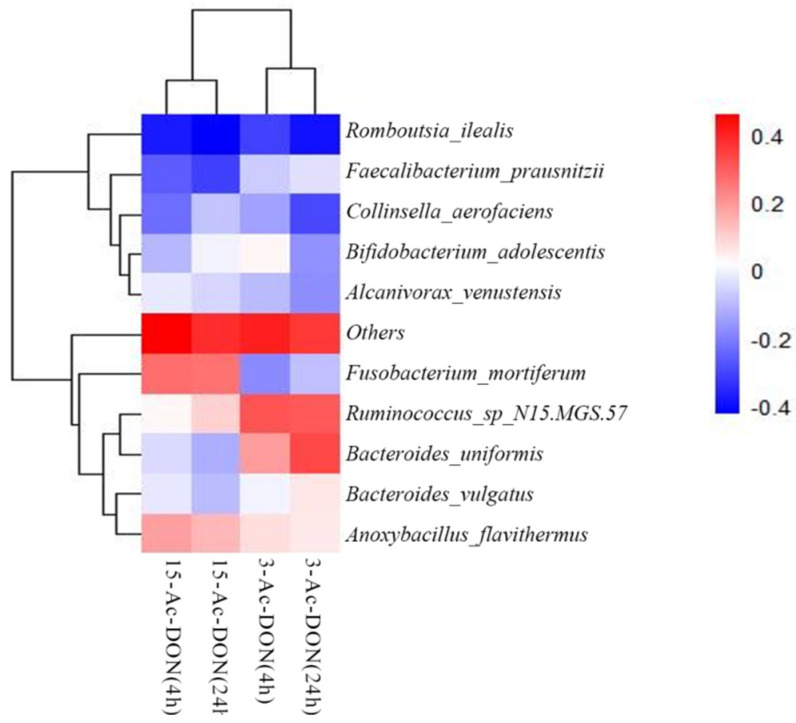
Spearman correlations between the taxon abundance at species level and the concentrations of acetylated DONs microbial metabolite present in incubations with fecal slurries from 25 individuals at 4 h and 24 h.

**Table 1 toxins-14-00199-t001:** Estimated transformation rate for the intestinal microbial formation of DON from 3-Ac-DON and 15-Ac-DON (4 h incubation, unless indicated otherwise).

	3-Ac-DON						15-Ac-DON						
Individuals	DON Concentration (μM)	Transformation Percentage (%)	Transformation Rate (In Vitro) ^a^	Calculated Transformation Time (min)	Reported Fecal Transit Time (min)	Conversion Rate (%) of Total Parent Compounds	DON Concentration (μM)		Transformation Percentage (%)	Transformation Rate (In Vitro) ^a^	Calculated Transformation Time (min)	Reported Fecal Transit Time (min)	Conversion Rate (%) of Total Parent Compounds
1	3.54 ± 0.20	1.8	0.44	452	1848	100	0.00	/	/	/	/	1848	/
2	18.24 ± 0.61	9.1	2.28	88		100	2.18 ± 0.38		1.1	0.27	734		100
3	23.56 ± 0.75	11.8	2.94	68		100	3.56 * ± 1.13		1.8	0.07	2698		68.5
4	35.40 ± 1.14	17.7	4.42	45		100	5.40 ± 1.97		2.7	0.67	296		100
5	16.49 ± 0.56	8.2	2.06	97		100	5.32 ± 0.89		2.7	0.66	301		100
6	6.52 ± 0.76	3.3	0.81	245		100	3.36 * ± 1.39		1.7	0.07	2859		64.6
7	8.48 ± 0.99	4.2	1.06	189		100	2.00 * ± 0.15		1.0	0.04	4805		38.5
8	10.56 ± 0.47	5.3	1.32	152		100	2.74 * ± 0.20		1.4	0.06	3508		52.7
9	25.97 ± 0.14	13.0	3.24	62		100	9.26 ± 0.00		4.6	1.16	173		100
10	10.52 ± 0.31	5.3	1.31	152		100	3.60 * ± 0.27		1.8	0.08	2665		69.3
11	21.29 ± 0.82	10.6	2.66	75		100	7.11 ± 0.80		3.6	0.89	225		100
12	3.71 ± 0.11	1.9	0.46	431		100	2.11 * ± 0.13		1.1	0.04	4546		40.6
13	5.42 ± 0.21	2.7	0.68	295		100	4.99 * ± 3.24		2.5	0.10	1925		96.0
14	13.87 ± 0.33	6.9	1.73	115		100	2.74 ± 0.10		1.4	0.34	584		100
15	6.53 ± 0.00	3.3	0.82	245		100	2.70 ± 0.59		1.3	0.34	593		100
16	9.23 ± 0.50	4.6	1.15	173		100	2.27 ± 0.16		1.1	0.28	705		100
17	13.87 ± 0.33	6.9	1.73	115		100	4.24 ± 0.03		2.1	0.53	377		100
18	27.96 ± 7.92	14.0	3.49	57		100	5.66 ± 0.41		2.8	0.71	283		100
19	12.97 ± 0.11	6.5	1.62	123		100	3.67 ± 0.23		1.8	0.46	437		100
20	6.68 ± 1.69	3.3	0.83	240		100	3.52 ± 0.24		1.8	0.44	455		100
21	3.71 ± 0.09	1.9	0.46	431		100	2.97 ± 0.24		1.5	0.37	540		100
22	27.27 ± 3.70	13.6	3.41	59		100	5.98 ± 0.27		3.0	0.75	268		100
23	8.64 ± 2.19	4.3	1.08	185		100	2.64 ± 0.32		1.3	0.33	606		100
24	11.12 ± 1.58	5.6	1.39	144		100	2.20 ± 0.01		1.1	0.27	728		100
25	14.42 ± 1.18	7.2	1.80	111		100	4.70 ± 0.87		2.4	0.59	341		100

^a^—nmol/min/g of content. /—no detected metabolites. *—The data obtained from 24 h incubation.

## Data Availability

The data presented in this study are available in this article and Appendix A.

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
