# Peer review of "Interindividual Differences in In Vitro Human Intestinal Microbial Conversion of 3-Acetyl-DON and 15-Acetyl-DON"

_toxins, 2022, doi:10.3390/toxins14030199_

Round 1

Reviewer 1 Report

A manuscript entitled: Interindividual differences in in vitro human intestinal microbial conversion of 3-acetyl-DON and 15-acetyl-DON was submitted for review. The work aimed to present potential interindividual differences in the conversion of acetylated DONs by intestinal bacteria and to evaluate the results in a risk assessment of acetylated DONs. Additionally, the aim was also to determine the correlation between kinetic parameters and taxon abundance in different human faecal samples. The work is interesting, however, I submit some comments and suggestions for its improvement:

Line 31: Fusarium graminearum - please replace "G" with a lowercase "g"

Line 44: Noted "(...) the gastrointestinal." shouldn't it be the gastrointestinal track?

Line 45: Please use the word consistently: gastro-intestinal or gastrointestinal?

Line 57: Please remove unnecessary spaces in the text

Line 68: It seems more reasonable to clarify the purpose of the paper as presenting the results of scientific inquiry on this topic.

Line 85: What symbols were used to designate the study samples?H1-H 25 or No. 1 - No. 25?Please use this choice consistently in the text of the paper, especially from Line 203 onwards.Please briefly justify the choice of volunteers for the study

Line 86: I recommend: Individual information is provided in the Supplementary Material Table 1.

Line 133: Replace uL with µL

Line 178: Figure 2- names of microorganisms replace with italic font

Line 180: Figure 3- the names of the micro-organisms are replaced by italic font

Line 205: What specifically refers to (8)? Please clarify the sentence so that it is clear

Line 215: Figure 6A- edit the table and move it to Figure 5A in the text of the paper

Lines 266-267 and 269: write the names of the microorganisms in italics

Line 332: please remove unnecessary spaces in the text

Line 345: write the names of the microorganisms in italic font

Line 433: Table 1 - edit table to make names in columns visible and clear by changing font size or text direction ect.)

Line 435: description under Table 1 is unclear, editing required

In my opinion, conclusions should be presented with a high degree of probability, with an indication of the need for in-depth research in this area. Please refine the literature according to the requirements of the journal.

Reviewer 2 Report

L6 : please clarify here that fecal microbiota was used (not necessary L9-10)

Clarify in the abstract that conversions of 13-Ac-DON and 15-Ac-DON were correlated

L31: Fusarium graminearum

L41: add some information of concentration of DON in food

L74: material and methods are usually at the end of the manuscript

Fig 2: please clarify this figure for people unfamiliar with this kind of representation (The numbers on the flower petals correspond to the genera common to the individuals? What do the colors correspond to?)

L191: why 200 µM of DON was used?

Discussion:

1- Please discuss the concentration of DON used regarding level of DON in food

L305-306- Please add a short paragraph on the representativeness of the fecal microbiota compared to the microbiota in the jejunum and ileon

Reviewer 3 Report

The paper is about the interindividual differences in converting 3AcDON and 15AcDON to DON in an in vitro human faecal system.

The paper is well written and contains all relevant information about the methodology. The results are presented correctly. The discussion of the results is mostly correct, but there are some hypothetical statements. In the conclusions, it would be important to add some further perspectives of these results other than the description of the findings.

The paper requires editorial correction.

Some remarks in order of appearance in the text:

L 16: faecal transit – please re-phrase this term because it is not faecal but chyme transit in the intestine

L 21: faecal transit – see my previous remark

L 44: gastrointestinal tract

L 44-45: Natural acute DON toxicity is rare in the human population; therefore, it would be important to know the acute toxic contamination level.

L 48-49: PEDV infection is not zoonotic; therefore, it has no relevance in humans.

L 79: Massachusetts is a state of USA. Please, add the city, as well.

L 80-81: Romer Labs (Tulln, Austria).

L 88: Are there any information about the volunteers' intake of any probiotic preparations?

L 93-95: Please describe the sterile faecal sample collection method, which is important for bacterial determinations.

L 137: California is a state of USA. Please, add the city, as well.

L 140-141: Please note the criteria for selecting primers.

L 154: What does it mean to control the group?

L 154: Please add the city and state.

L 201-209: There is no information about the presence of other metabolites by using 15AcDON.

L 236: Faecal transit time – see my previous remarks about this term.

L 251-253: The hydrolysis of the acetylated DON metabolites did not occur in the faecal samples but during the transit through the gastrointestinal tract. However, there is no information about which part of the GIT has importance for this process. It would be important because the absorption of DON from different parts is different, and the large intestine is less possible.

L 287-296: This paragraph is correct, but it would also be important to know about the absorption of acetylated DON metabolites from the small intestine.
